# Role of Alternative Elicitor Transporters in the Onset of Plant Host Colonization by *Streptomyces scabiei* 87-22

**DOI:** 10.3390/biology12020234

**Published:** 2023-02-01

**Authors:** Isolde M. Francis, Danica Bergin, Benoit Deflandre, Sagar Gupta, Joren J. C. Salazar, Richard Villagrana, Nudzejma Stulanovic, Silvia Ribeiro Monteiro, Frédéric Kerff, Rosemary Loria, Sébastien Rigali

**Affiliations:** 1Department of Biology, California State University, Bakersfield, CA 93311-1022, USA; 2InBioS-Center for Protein Engineering, University of Liège, Institut de Chimie, B6a, B-4000 Liège, Belgium; 3Department of Plant Pathology, University of Florida, Gainesville, FL 32611-0180, USA

**Keywords:** *Streptomyces scabiei*, cello-oligosaccharide, sugar transport, virulence, plant pathogen, elicitor sensing, phytotoxin, virulome, cellulose utilization, secondary metabolism

## Abstract

**Simple Summary:**

The bacterium *Streptomyces scabiei* is the main causative agent of common scab disease on economically important root and tuber crops. Our work investigated if and how this pathogen uses multiple sugar transport systems to sense the presence of a living plant host through perception and import of cello-oligosaccharides, the elicitors that trigger the onset of the pathogenic lifestyle of *S. scabiei*.

**Abstract:**

Plant colonization by *Streptomyces scabiei*, the main cause of common scab disease on root and tuber crops, is triggered by cello-oligosaccharides, cellotriose being the most efficient elicitor. The import of cello-oligosaccharides via the ATP-binding cassette (ABC) transporter CebEFG-MsiK induces the production of thaxtomin phytotoxins, the central virulence determinants of this species, as well as many other metabolites that compose the ‘virulome’ of *S*. *scabiei*. Homology searches revealed paralogues of the CebEFG proteins, encoded by the *cebEFG2* cluster, while another ABC-type transporter, PitEFG, is encoded on the pathogenicity island (PAI). We investigated the gene expression of these candidate alternative elicitor importers in *S*. *scabiei* 87-22 upon cello-oligosaccharide supply by transcriptomic analysis, which revealed that *cebEFG2* expression is highly activated by both cellobiose and cellotriose, while *pitEFG* expression was barely induced. Accordingly, deletion of *pitE* had no impact on virulence and thaxtomin production under the conditions tested, while the deletion of *cebEFG2* reduced virulence and thaxtomin production, though not as strong as the mutants of the main cello-oligosaccharide transporter *cebEFG1*. Our results thus suggest that both *ceb* clusters participate, at different levels, in importing the virulence elicitors, while PitEFG plays no role in this process under the conditions tested. Interestingly, under more complex culture conditions, the addition of cellobiose restored thaxtomin production when both *ceb* clusters were disabled, suggesting the existence of an additional mechanism that is involved in sensing or importing the elicitor of the onset of the pathogenic lifestyle of *S*. *scabiei*.

## 1. Introduction

In order to optimally benefit from opportunities that will enhance their competition, survival, and spread, soil borne phytopathogens need to be able to respond adequately to their environment. Hereby, it is crucial that phytopathogens can distinguish common soil elements from signals indicating the presence of a plant host for the onset of their virulence mechanisms. Plant cell wall components such as cellulose, xylan, and their respective oligosaccharide degradation products are common triggers for saprophytic and plant pathogenic microorganisms [1]. Pathogenicity by the plant pathogenic streptomycetes *Streptomyces scabiei*, *S. acidiscabies*, and *S. turgidiscabies*, the main causative agents of common scab disease on various economically important root and tuber crops [2], is also triggered by plant cell wall components [3], more specifically, cellobiose and cellotriose are known to induce the production of their main virulence factor, thaxtomin A [3,4]. This phytotoxin targets the cellulose synthase enzyme that is active in expanding and dividing plant cell tissue, leading to stunted plant growth, cell hypertrophy, and tissue necrosis [5,6]. Apart from thaxtomin phytotoxins, cellobiose and particularly cellotriose, also induce the production of many other specialized metabolites, such as concanamycin phytotoxins, siderophores to acquire iron for housekeeping functions, ectoine, possibly for protection against osmotic shock once inside the host, and bottromycin and concanamycin antimicrobials, most likely to prevent other microorganisms from colonizing the same niche [7].

Considering that cellulose is the most abundant polysaccharide on earth, this adaptation, however, comes with the difficulty of distinguishing between cello-oligosaccharides originating from the saprophytic degradation of dead plant material and cello-oligosaccharides indicating the presence of a nearby living host. We hypothesized previously that the ability of *S*. *scabiei* to discriminate between dead and living plants to initiate its main virulence strategy would in part reside in the fact that this strain has disabled its cellulolytic system and therefore cannot generate cellobiose, which is by far the major byproduct released by cellulases on dead plant cell walls [8]. On the other hand, cellotriose (and not cellobiose) has been shown to emanate from living plants upon the action of thaxtomins [4]. Cellotriose, which is also the best elicitor of the virulome [7], would therefore signal the presence of a living host to colonize while cellobiose can only be generated by the soil microbial community [8].

The main ATP-binding cassette (ABC) transporter responsible for actively importing cellobiose and cellotriose has been identified [9]. It consists of CebE as a cello-oligosaccharide-binding protein, CebF and CebG as membrane proteins that form the permease, and MsiK, the ATPase that provides the energy for this active transport. The imported sugars are then hydrolyzed by the beta-glucosidase BglC to feed the glycolysis with glucose [10]. Importantly, protein–ligand interaction studies revealed that the CebE protein of *S*. *scabiei* possesses K_D_ values in the nanomolar range towards cellotriose and cellobiose, whereas the CebE protein of the highly cellulolytic strain *S. reticuli* displays K_D_ values in the micromolar range [9]. The higher affinity of CebE of *S*. *scabiei* to cello-oligosaccharides and especially to cellotriose could be a key adaptation for this bacterium to perceive this molecule as a signal instead of a nutrient and trigger appropriate responses such as thaxtomin production. Cellobiose and cellotriose induce the production of thaxtomin, not only through interaction with the pathway-specific activator of the thaxtomin biosynthetic cluster, TxtR [11], but also mainly by relieving inhibition of *txtR* transcription by the cellulose utilization regulator CebR [12]. Indeed, plant pathogenic streptomycetes seem to have evolved to recruit a regulator associated with primary metabolism in nonpathogenic *Streptomyces* species as the major gatekeeper of the production of their most important virulence factor. Interestingly, except for thaxtomins, none of the biosynthetic gene clusters involved in the production of the specialized metabolites induced by the import of cello-oligosaccharides is under direct control of CebR, suggesting the existence of a yet unknown mechanism for switching on the other components of the virulome [7].

In an earlier study, we observed the unexpected survival of the deletion mutant of *bglC* on minimal medium supplied with cellobiose as a unique carbon source, revealing that *S*. *scabiei* 87-22 possesses at least one alternative beta-glucosidase (BcpE1) that resides in a second *cebR*-*cebEFG* cluster [13]. In addition, genes encoding an ABC-type sugar transport system are also found in the pathogenicity island (PAI) of thaxtomin-producing pathogenic *Streptomyces* species [9,14]. In this study, we evaluated the potential role of these two candidate alternative cello-oligosaccharide importers in the onset of the pathogenic lifestyle of the common scab causing model species *S. scabiei* 87-22.

## 2. Materials and Methods

### 2.1. Bacterial Strains, Media, and Culture Conditions

All strains and plasmids used in this study are described in Table A1. *Escherichia coli* strains were cultured in Luria–Bertani (LB) medium at 37 °C. *Streptomyces* strains were routinely grown at 28 °C in Tryptic Soy Broth (TSB) or on International *Streptomyces* Project medium 4 (ISP-4, BD Biosciences, Franklin Lakes, NJ, USA). When required, the medium was supplemented with the antibiotics apramycin (100 µg/mL), kanamycin (50 µg/mL), chloramphenicol (25 µg/mL), thiostrepton (25 µg/mL), spectinomycin (50 µg/mL), streptomycin (100 µg/mL), and/or nalidixic acid (50 µg/mL).

Mycelial suspensions for the different assays were prepared by growing the strains as precultures in 5 mL TSB for 2 days at 28 °C while shaking at 220 rpm. Pellets were harvested and washed twice with sterile distilled water and resuspended in sterile distilled water to reach an optical density of 1.0 at 600 nm (OD_600_).

### 2.2. In Silico Analysis

A search for the orthologues of SCAB57751 (CebE1) [9], SCAB2421 (CebE2), and SCAB77271 (PitE) in various pathogenic *Streptomyces* species was performed via the online BLASTP tool [15]. Retrieved amino acid sequences were aligned and phylogenetic trees were constructed through the ‘Advanced’ workflow of NGPhylogeny.fr (https://ngphylogeny.fr/workflows/advanced/fastme-oneclick? accessed on 23 January 2023) with all parameters and methods set by default [16] with the addition of the option ‘Bootstrap branch supports’. Computational prediction of the binding sites of CebR2 was performed using the PREDetector software available at http://predetector.hedera22.com/ [17], with a training set of sequences obtained via the AURTHO approach [18]. The in silico analyzed strains and genome accession numbers are listed in Table A2.

### 2.3. Transcriptomics

*S*. *scabiei* 87-22 cultivation, RNA preparation, and RNA-seq mapping and analyses were performed as previously described [7]. RNAseq data were publicly deposited, and our experimental and analytical pipelines are described in the GEO database repository (accession number: GSE181490). Values (Log2 fold-change) express the transcriptional response after 1 h and 2 h post addition of cellobiose or cellotriose. Genes of the thaxtomin biosynthetic cluster were used as positive controls, whereas genes of housekeeping function and of other sugar transporters were used as negative controls.

### 2.4. Construction of Deletion Mutants

Deletion mutants in *S. scabiei* 87-22 were created using the REDIRECT^©^ PCR targeting methodology [19] that replaced the gene(s) of interest by an antibiotic resistance deletion cassette, as described previously [12]. Double mutants were generated using deletion cassettes conferring resistance to different antibiotics. The primers used to generate the deletion cassettes and verification of the created gene deletions are listed in Table A3. The protocol for genomic DNA extraction is detailed in Appendix B.

### 2.5. Analysis of Thaxtomin Production

For visual comparison of growth phenotypes and quantification of thaxtomin production, small agar plates (35 mm diameter, 5 mL) with either TDM (thaxtomin defined medium) [4] supplemented with 1% cellobiose or OBA (oat bran agar) [4], with or without 1% cellobiose, were inoculated with 40 µL of a mycelial suspension at OD_600_ 1.0 (mycelia of 48 h TSB cultures, washed twice in distilled water) of the *S. scabiei* 87-22 wild-type and the deletion mutants and incubated for at 28 °C. Plates were evaluated and thaxtomin was extracted from the agar plates at 3 and 7 days post inoculation (dpi). Thaxtomin extraction and HPLC analysis of the samples were performed as described previously [9]. The assays were performed at least two times with two technical repeats of each strain and with at least two isolates of each mutant.

### 2.6. Virulence Assays

The virulence phenotype of the mutant isolates was evaluated through in vitro radish seedling assays as described previously [9]. The assays were performed three times with at least two isolates of each mutant.

### 2.7. Modeling of CebE2

The 3D model of CebE2 was generated using AlphaFold v2.3.0 [20], as implemented in the AlphaFold Colab notebook (https://colab.research.google.com/github/deepmind/alphafold/blob/main/notebooks/AlphaFold.ipynb#scrollTo=pc5-mbsX9PZC accessed on 30 December 2022).

## 3. Results

### 3.1. In Silico Analysis of CebE Homologs of Streptomyces scabiei

In our initial search for the cellobiose/cellotriose transporter of *S. scabiei* based on sequence homology with the proteins of the known cellobiose ABC-type transporter of *S. reticuli* (CebE^reti^) [21] and of *S*. *griseus* (CebE^gris^) [22], three gene clusters were identified, which is in accordance with what was mentioned in previous reports [9,11,14]. The cluster with the highest homology to both CebE^reti^ and CebE^gris^ (*scab57761-57721*, Figure 1) was previously demonstrated to code for the high-affinity cellobiose/cellotriose-specific CebEFG-MsiK transporter of *S. scabiei* [9]. This gene cluster also contains the gene for the transcriptional repressor CebR that controls both cello-oligosaccharide utilization and thaxtomin production [12], and the gene encoding the beta-glucosidase BglC for cello-oligosaccharide catabolism [10]. This first *cebR*-*cebEFG*-*bglC* gene locus is highly conserved among pathogenic streptomycetes, including the species *S. acidiscabies*, *S. ipomoeae*, *S. stelliscabiei*, *S. griseiscabiei*, *S. brasiliscabiei*, *S. europaeiscabiei*, and *S. niveiscabiei* (Figure 2a). The CebEs of pathogenic strains belonging to the species *S. caniscabiei*, *S. reticuliscabiei*, *S. turgidiscabies*, and *S. puniscabiei* are phylogenetically related and grouped with the CebE^reti^ (Figure 2a). It has to be noted that none of the isolated pathogenic species possess a CebE protein that is orthologous to CebE^gris^ (Figure 2a).

The second gene cluster that was identified contains the paralogues (from now on called *cebR2*, *cebE2*, *cebF2*, *cebG2, and bglC2/bcpE1*) of the first *ceb* gene locus with identical synteny (*scab2391-2431,*
Figure 1). The putative role of this second *ceb* cluster in cello-oligosaccharide transport and utilization is supported by the recent biochemical characterization of the gene encoding the beta-glucosidase BcpE1 (BlgC2), which displayed similar kinetic properties for cellobiose hydrolysis and that was proven to compensate for the genetic loss of *bglC* of the first *ceb* cluster [13]. The search for *cis*-acting elements in the upstream region of *scab2421* (*cebE2*) identified two sequences (TGGGAACGTgCCCA and TGGGcACGTTCCCA) that displayed only one mismatch (at symmetrical positions 9 and 6, respectively) with the 14 nt-long TGGGAACGTTCCCA palindromic consensus sequence deduced for CebR2 proteins via the AURTHO approach [18]. This high similarity between CebR (TGGGACGCGTCCCA) and CebR2 *cis*-acting elements further supports the hypothesis that these *ceb* clusters have originated from a common ancestor. Surprisingly, the 19 pathogenic strains that possess this putative second cello-oligosaccharide transporter all belong to the species *S. scabiei* (Figure 2a, Table A2).

Finally, a third sugar ABC-type transporter with partial homology, *scab77271*-*77231*, is located within the pathogenicity island (PAI) and has been identified among all the thaxtomin A-producing pathogenic *Streptomyces*, except for strains belonging to *S. ipomoeae* (Figure 2b, Table A2). From now on, genes of this third transporter will be called *pitE*, *pitF*, and *pitG*, with *pit* standing for pathogenicity island transporter. Importantly, neither CebR- nor CebR2-binding sites were identified upstream of *pitE* (*scab77271*), suggesting that the expression of the *pit* cluster responds to other environmental elicitors.

### 3.2. Transcriptional Response of Alternative Cello-Oligosaccharide Transporters to Cellobiose and Cellotriose

To evaluate if these two alternative transporters are potentially also involved in the import of cello-oligosaccharide elicitors by *S*. *scabiei* 87-22, we assessed their expression response post addition of cellobiose and cellotriose. RNA samples were collected 1 and 2 h post-addition of each elicitor [7]. As shown in Figure 3, the expression of the genes of the second *ceb* operon (*cebE2*, *cebF2*, *cebG2*, and *bcpE1*/*bglC2*) was drastically induced by cellobiose, with an average Log2-fold change (LFC) of 4.6 and 5.75 at 1 and 2 h post-addition of cellobiose, respectively. These values are in the same order of magnitude, although somewhat lower, as observed for the first *ceb* operon with an average LFC of 7.5 and 7.4. The genes of the *pitEFG* transcription unit showed an average LFC of 1.0 and 2.3 at 1 and 2 h post-addition of cellobiose, respectively. Although these fold changes suggest a significant response compared to the negative control genes (1.0 LFC as threshold fixed for a significant response), they are much lower than what is observed for the other two transporters, and for the genes of the thaxtomin biosynthetic gene cluster (positive controls) (Figure 3). Regarding the effect of cellotriose, gene expression of *cebE2*, *cebF2*, and *cebG2* displayed an average LFC of 1.7 and 2.8 at 1 and 2 h post-addition of cellotriose, respectively. These values are weaker than those observed for the first transporter, with an average LFC of 4.8 and 5.9, but still representing a highly significant upregulation. The expression of the three transporter genes of the *pit* operon did not show a significant response with an average LFC of 0.55 and 0.87 at 1 and 2 h post-addition of cellotriose, respectively. Interestingly, the gene encoding the predicted beta-glucosidase (*scab_77241*) of the *pit* cluster showed a stronger expression response to cellotriose compared to the rest of the predicted transcription unit, with an LCF of 3.19 at 2 h post-addition of the elicitor.

Overall, our transcriptomic study revealed that the genes encoding a putative alternative transporter of cello-oligosaccharides (*cebEFG2*) are significantly transcriptionally activated by the presence of cellobiose and cellotriose, whereas the expression response of the genes of the transporter located on the pathogenicity island is much weaker, barely significant. These transcriptomic data, post-elicitor induction, are in line with earlier qRT-PCR results performed on RNA samples collected after 24, 48, and 72 h of growth of *S*. *scabiei* EF-35 in minimal medium supplemented with cellobiose (0.5%) [14].

### 3.3. Role of the CebEFG2 and PitEFG Transporters in Virulence Elicitor Utilization by Streptomyces scabiei

Specific deletion mutants in the genes encoding either the solute-binding proteins and/or all three proteins of the transporter were created to evaluate the role of the homologous transporters for growth on cellobiose and in the *S. scabiei* pathology. Figure 4 shows the ability of the various *ceb* and *pit* gene cluster deletion mutants to grow on minimal medium supplemented with cellobiose as the unique carbon source (TDMc). The differences compared to the *S*. *scabiei* wild-type strain (87-22) are optimally observed at three days post-inoculation, where the *cebEFG1* deletion mutant (Δ*cebEFG1*) showed a drastic growth delay, which is partially made up at seven days post-inoculation (Figure 4). The restored growth of mutant Δ*cebEFG1* at seven days post-inoculation is most likely due to the strong activation of the second *ceb* cluster by cello-oligosaccharides (Figure 3), and which also codes for a beta-glucosidase BcpE1 (BglC2) that possesses hydrolytic properties towards cellobiose similar to the BglC connected to CebEFG1 [13]. Instead, deletion of either the transporter encoding genes of the second *ceb* cluster (Δ*cebEFG2*) or the sugar-binding protein of the pathogenicity island transporter (Δ*pitE*) did not significantly affect growth on TDMc. The double mutant for both *cebE* genes (Δ*cebE1-2*) was even more impaired in growth on TDMc, further confirming that both transporters, CebEFG1 and CebEFG2, participate in concert in cellobiose uptake.

Upon inoculation of radish seedlings, *S. scabiei* typically causes strong root and shoot stunting, swelling of the hypocotyl, and necrosis of the root tip (Figure 5a). Disabling of either one or both *ceb* clusters resulted in less virulent strains. Seedlings inoculated with the mutants were able to partially develop compared to the wild type 87-22 (Figure 5a). This attenuated virulence phenotype is most likely due to reduced thaxtomin production, measured when the toxin was extracted from the agar–water medium the plants were grown in (Figure 5b). Despite the conservation and localization of the *pitEFG* transporter genes within the PAI of the pathogenic streptomycetes, the loss of its solute-binding protein did not seem to affect the virulence capacity under the conditions tested (Figure 5).

As displayed in Figure 6, reduced toxin production was also measured when growing the *ceb* mutants on OBA, a plant-based medium known to induce thaxtomin production [4]. Thaxtomin production levels were, however, restored when cellobiose was added to the medium (OBAc), even for the mutant strains where both cellobiose transporters were disabled (Figure 6). These results confirm the observations for the reduced virulence capacity of the mutants when inoculated onto radish seedlings and that under more complex conditions, as there would be in plant–microbe interactions, the loss of the CebEFG2 transporter cannot be entirely compensated for by the CebEFG1 transporter.

## 4. Discussion

In this work, we showed that *S. scabiei*, the most widespread and oldest potato scab-causing pathogen [23], harbors paralogues to its main cello-oligosaccharide transporter that was previously identified (CebEFG1) [9]. Expression of *cebEFG2*, encoding the transporter with the highest homology to CebEFG1, responds to the same main cello-oligosaccharide virulence elicitors, cellobiose and cellotriose. This is in line with qRT-PCR results from earlier work on *S. scabiei* EF-35 [14] and indicates that this transporter is indeed an alternative to CebEFG1 and therefore could serve as a safeguard for the loss of the main transporter. However, when grown on the complex plant-based medium OBA or during plant–microbe interaction, CebEFG2 seems to have a significant role.

Considering that cellulose is the most abundant polysaccharide on earth, the uptake and consumption of cello-oligosaccharides was anticipated to be a key feature for soil-dwelling streptomycetes. Hence, it is surprising that multiple gene loci exist that are dedicated to this specific function with at least four different *ceb* gene clusters, namely the main *cebR-cebEFG-bglC* gene clusters of *S. scabiei* [9], *S. reticuli* [21,24], and *S. griseus* [22], and the paralogous *cebR2-cebEFG2-bcpE1/bglC2* gene cluster found in most *S. scabiei* species. These three main types of *ceb* clusters are not orthologous as they differ from one another by (i) their low level of homology (the percentages of amino acid identity between CebE proteins are below 50%, much lower than the values usually found for orthologous genes/proteins), (ii) their gene organization (*cebR* is the last gene of the transcriptional unit in *S*. *griseus*, while it is divergently transcribed in *S*. *scabiei* and *S*. *reticuli*), and (iii) their affinity for binding cellobiose and cellotriose (at the nano- and micromolar levels for CebE^scab^ and CebE^reti^, respectively) [9,21]. Importantly, the molecular mechanism governing the expression control of these genes has been conserved as the transcriptional CebR repressors of *S. scabiei*, *S. reticuli*, and *S. griseus* all recognize the same *cis*-acting element with the palindromic TGGGACGCGTCCCA as consensus sequence [12,22,24], which allows them to also control the transcription of the cellulase encoding genes [25].

The second and alternative cello-oligosaccharide gene cluster (*cebEFG2*) seems to only be present in the *S*. *scabiei* group of pathogenic *Streptomyces* species. Even the phylogenetically closest pathogenic strains to the *S. scabiei* group, such as *S*. *ipomoeae*, *S*. *brasiliscabiei*, and *S*. *griseoscabiei*, do not possess this second cello-oligosaccharide transporter (CebEFG2^scab^). This CebEFG2^scab^ transporter is not exclusive to pathogenic *S. scabiei* species as it is also found in diverse non-pathogenic *Streptomyces* species, including *S. coelicolor* [26] and *S. lividans* [27] as the best-studied species, amongst many others according to BlastP results (not shown). The question remains whether or not other pathogenic streptomycetes that have a CebE^scab^ (species *S. acidiscabies, S. ipomoeae, S. europaeiscabiei, S. niviscabiei, S. stelliscabiei, S. brasiliscabiei,* and *S. griseiscabiei*) or a CebE^reti^ (*S. turgidiscabies, S. caniscabiei*, *S. reticuliscabiei*, and *S. puniscabiei*) background potentially possess an additional and yet unidentified *ceb* cluster for cello-oligosaccharide import and/or to safeguard the possible loss of the main *ceb* cluster. Another intriguing observation is that the PAI that contains the genes essential for plant host colonization has never been transferred and integrated into the chromosome of *Streptomyces* species that possess a CebE^gris^ background. Why this has never occurred remains unexplained. Maybe these species lack other genetic features that are indirectly essential for plant host colonization, which would minimize the selection pressure to keep the PAI if integrated in these species, unless scab-causing *Streptomyces* species containing CebE^gris^ have not been isolated yet.

The reason why CebEFG2 is less efficient than the main CebEFG1 transporter at both using cello-oligosaccharides as a nutrient source or at importing the virulence elicitors remains to be studied at the molecular level. We recently obtained the crystallographic structure of CebE1^scab^ in complex with cellotriose and are currently investigating the amino acids involved in sugar recognition (publication in preparation). Based on the crystallographic structure of CebE1^scab^, we generated the 3D model of CebE2 using AlphaFold v2.3.0 [20], as implemented in the AlphaFold Colab notebook. The CebE2 model is characterized by an overall good quality with an average pLDDT value of 95.9 for the folded region (aa 23-425). This model corresponds to an open conformation of the protein. However, when superimposed separately, each domain is similar to the structure of CebE1 in complex with cellotriose, with rms deviation of 1.01 Å over 135 Cα and 0.93 Å over 190 Cα for Domain 1 and 2, respectively (Figure 7a). The opening of the ligand binding pocket corresponds to an approximate 45° rotation of Domain 2 compared to Domain 1 (Figure 7b). According to the structural model of CebE2 (Figure 7), Domain 1 interacts with the ligand through seven hydrogen bonds (H-bonds) that are conserved between CebE1 and CebE2 (sidechain of D143 and W302; mainchain of V42, E71, Q72, and G304), and 3 hydrophobic interactions, of which one is conserved (F40) and two are equivalent (N101 sidechain in CebE1 vs. Q71 in CebE2; the A73-M123 pair in CebE1 replaced by the M43-G93 pair in CebE2) (Figure 7c). In Domain 2 (Figure 7d), two major hydrophobic interactions are conserved in between CebE1 and CebE2 (F253 and W274) as well as two H-bonds involving the T424 side chain in CebE1 (S396 in CebE2). For the H-bonds involving the sidechains of Y307 and D425, the model does not establish whether H278 and Q397 can be substituent. The interaction between CebE1 and cellotriose is also mediated by a network of eight water molecules, of which four are fully conserved and the four are only partially conserved. Overall, because of the large number of interactions conserved, CebE2 is clearly compatible with the binding of cellotriose and cellobiose, such as CebE1. It is, however, difficult to extrapolate a difference in binding affinities between CebE1 and CebE2 for these ligands, as well as different import efficiencies of the entire machinery, and therefore how they individually contribute to informing *S. scabiei* of the presence of the inducer of its pathogenic lifestyle.

Another major difference between the *ceb1* and *ceb2* clusters of *S*. *scabiei* is the sequence of the consensus *cis*-acting elements recognized by CebR1 (TGGGACGCGTCCCA) and CebR2 (TGGGAACGTTCCCA), respectively. Although these two CebR-binding sites only differ in two symmetrical positions of their 14-nt inverted repeat consensus sequence, these nucleotide replacements may result in different binding affinities, which would ultimately lead to a different transcriptional response in terms of concentration of the allosteric effector required to dissociate the protein-DNA complex. This small modification also suggests that only CebR1 directly controls the thaxtomin biosynthetic gene cluster and cellulase encoding genes as the computational prediction of the CebR2 regulon seems to be strictly limited to the intergenic region between *cebR2* and *cebEFG2*-*bcpE1*/*bglC2*.

Perhaps the most surprising result of our study is the conserved ability of cellobiose to induce thaxtomin production in mutants where both cello-oligosaccharide transporters were disabled (strain Δ*cebE1-2*) when they are inoculated on OBAc (Figure 6, top right panel). This was unexpected as it therefore strongly suggests the existence of another mechanism to either import or sense cello-oligosaccharides in *S. scabiei* 87-22. We previously hypothesized the existence of another regulator than CebR able to sense and respond to cellobiose and cellotriose as, except for the thaxtomin, all other metabolites that are a part of the virulome of *S. scabiei* 87-22 have their expression induced by both elicitors despite the absence of CebR *cis*-acting elements in their biosynthetic gene clusters [7]. The identity of this additional transcriptional regulator of cello-oligosaccharide utilization is unknown and could be part of the numerous regulatory protein encoding genes that display a strong response to both cellobiose and cellotriose, as observed in our transcriptomic analysis [7].

Finally, we demonstrated here that genes coding for the PitEFG ABC-transporter that are located on the PAI would play no role in the onset of pathogenicity of *S. scabiei* 87-22 (Figure 5). This result raises the question why these sugar-transporter genes are systematically maintained on the PAI if they do not play a role in plant colonization. This is maybe due to the experimental set up of plant bioassays where the controlled conditions in which the radish seedlings grow do not generate the panel of carbohydrates that common scab causing *Streptomyces* species encounter in the rhizosphere. Once the substrate of PitE is identified, it will be easier to deduce or propose a role for this transporter in plant colonization.

## 5. Conclusions

In this work, we refined the proposed signaling pathway from cello-oligosaccharide transport to the production of thaxtomin and the onset of the pathogenic lifestyle of *S. scabiei* 87-22 (Figure 8). In addition to the main CebEFG-MsiK transporter, cello-oligosaccharides, can be imported by the alternative CebEFG2 that also recruits MsiK as ATPase as deletion of *msiK* fully abolished growth on TDMc [9]. The imported sugars are both hydrolyzed by beta-glucosidase BglC and BcpE1/BglC2 to feed the glycolysis with glucose and serve as allosteric effectors to inhibit the transcriptional repression of *txtR* by CebR, the pathway-specific activator of the thaxtomin biosynthetic genes. As cellobiose is still able to induce thaxtomin in OBAc when both *ceb* clusters are disabled (Figure 6, top right panel), we suggest the existence of an additional importer or sensor of cello-oligosaccharides that would directly or indirectly (via a yet unknown regulator) induce thaxtomin production. The role and the substrate specificity of the pathogenicity island-associated transporter PitEFG is still unknown.

## Figures and Tables

**Figure 1 biology-12-00234-f001:**
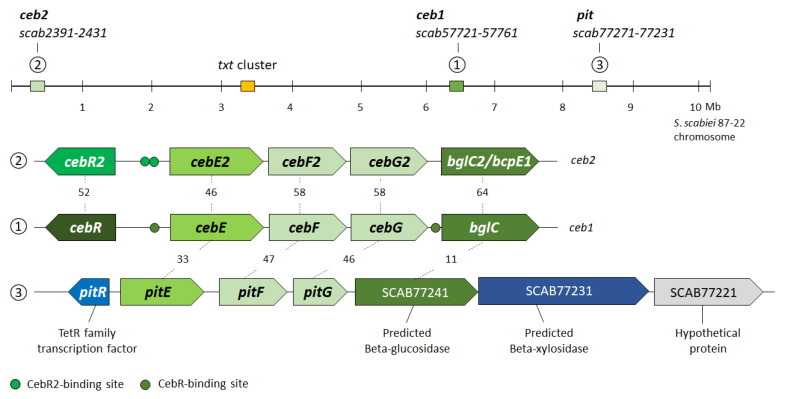
Chromosome position, gene organization, and identity levels of the three sugar ABC-type transporters presumed to interact with cello-oligosaccharide elicitors in *S*. *scabiei* 87-22. Numbers between ORFs are the percentage of amino acid identity between the corresponding homologous proteins.

**Figure 2 biology-12-00234-f002:**
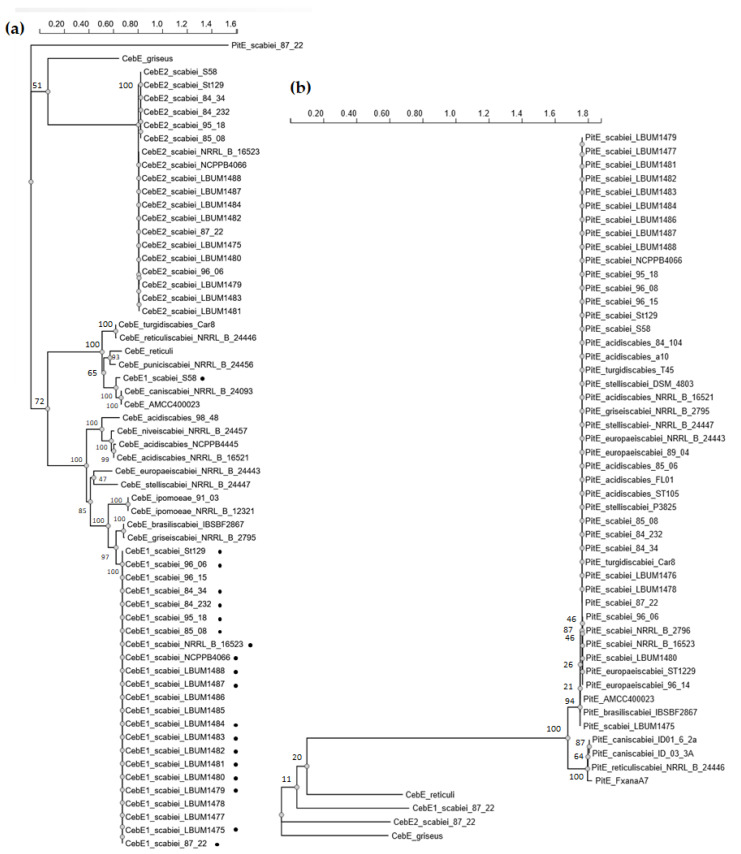
Phylogenetic analysis of CebE and PitE homologues. (**a**) Phylogram with CebE1 and CebE2 homologues. Black dots indicate the strains that possess both CebE1 and CebE2 proteins. Except for *S*. *scabiei* strains, only one representative of proteins with identical accession numbers was used to generate the tree; (**b**) Phylogram of PitE homologues. Bootstrap values of main nodes are indicated. The scale bar indicates the number of amino acid sequence substitutions per site. All accession numbers of proteins used to generate the phylograms are listed in Table A2 in Appendix A.

**Figure 3 biology-12-00234-f003:**
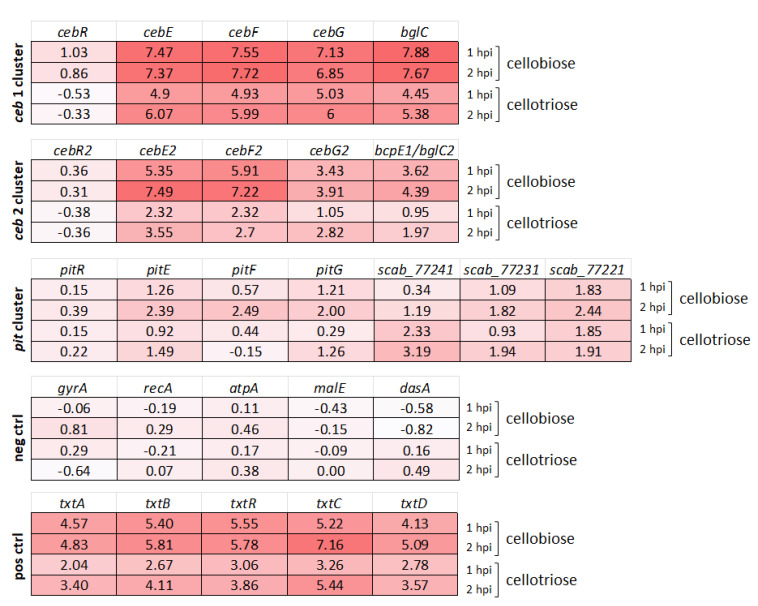
Transcriptional response of known and putative elicitor transporter clusters upon cellobiose and cellotriose supply. Values in the heatmap are the mean of Log2 fold-change. The expression changes of housekeeping genes *gyrA*, *recA*, *atpA*, and of *bxlE* and *dasA* encoding the xylobiose and *N*-N’-diacetylchitobiose ABC-type sugar-binding proteins, respectively, were used as negative controls (for which the expression is known to be independent of either cellobiose or cellotriose). The expression changes of five genes of the thaxtomin biosynthetic gene cluster (*txt*) were used as positive controls. abbreviations: neg ctrl, negative controls; pos ctrl, positive controls; hpi, hour(s) post induction by either cellobiose or cellotriose.

**Figure 4 biology-12-00234-f004:**
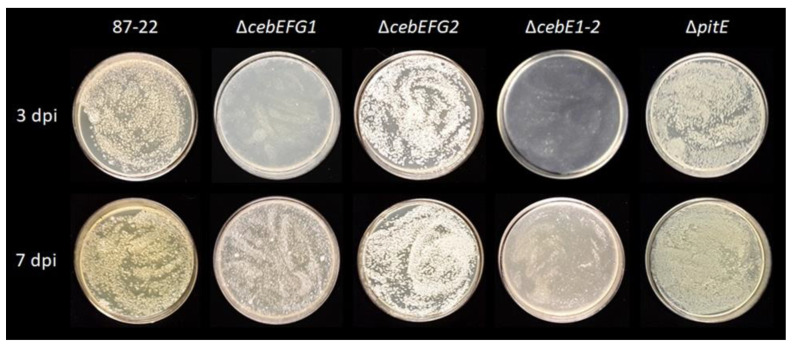
Growth of wild type (87-22) and transporter deletion mutant strains on minimal medium supplemented with cellobiose as the sole carbon source at 3 and 7 dpi. Note that the growth delay was only observed when genes of the *ceb*1 cluster are deleted (mutants Δ*cebEFG1* and Δ*cebE1-2).*

**Figure 5 biology-12-00234-f005:**
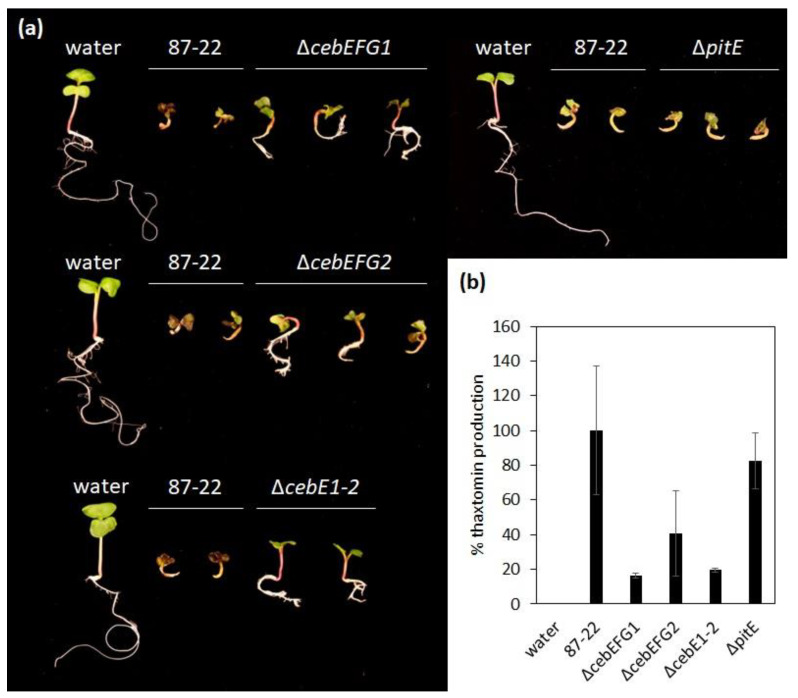
(**a**) Radish seedling assays showing the reduced virulence of the transporter deletion mutant strains compared to the wild type (87-22) at 7 dpi; (**b**) Thaxtomin levels as extracted from the radish seedling assay at 7 dpi and analyzed by HPLC. Thaxtomin levels produced by the wild type were set as 100%.

**Figure 6 biology-12-00234-f006:**
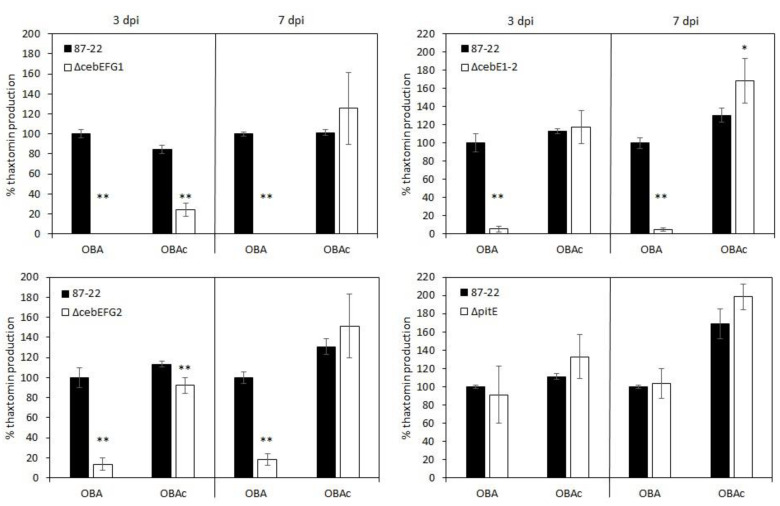
Thaxtomin production levels of the mutant strains compared to the wild type (87-22) on a plant-based medium without (OBA) and with (OBAc) the addition of cellobiose. Thaxtomin was extracted from agar plates at 3 and 7 dpi and analyzed by HPLC. Thaxtomin levels produced by the wild type were set as 100%. For the mutant strains, the means (± standard deviation) of at least three independently generated mutant isolates plated in duplicate are shown. Mutants that produced statistically different thaxtomin levels compared to the wild type grown under the same conditions are indicated by * if *p* < 0.05 and ** if *p* < 0.025.

**Figure 7 biology-12-00234-f007:**
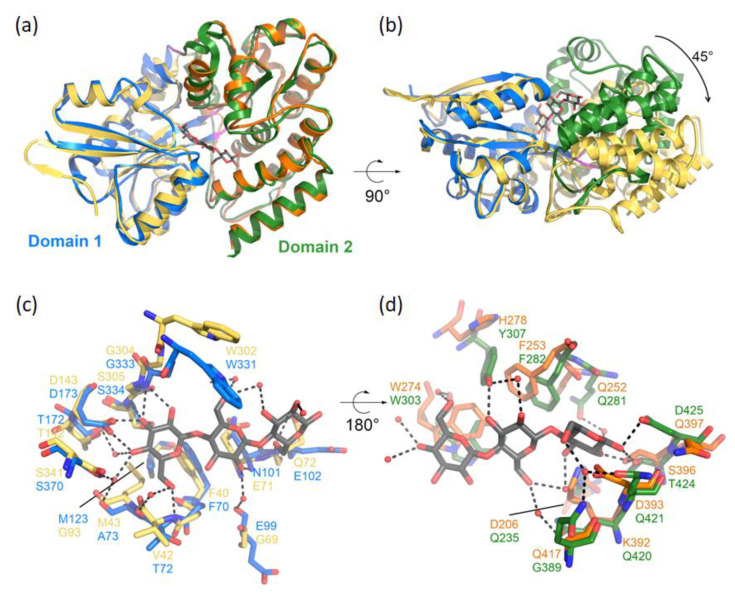
3D model of CebE2. (**a**) Cartoon representing the CebE1 (Domain 1 in blue and Domain 2 in green) structure in complex with cellotriose (grey sticks) with the two domains of CebE2 superimposed separately (Domain 1 in yellow and Domain 2 in orange). (**b**) CebE1 (same coloring scheme as in (**a**)) and CebE2 (yellow) with their Domain 1 superimposed, highlighting the 45° rotation observed between Domain 2 of both proteins. (**c**) Interactions (black dashed lines) between cellotriose (grey sticks) and Domain 1 of CebE1 (blue sticks), including through water molecules (red spheres). Equivalent residues in CebE2 are displayed as yellow sticks. (**d**) Same as (**c**) for Domain 2 with CebE1 in green and CebE2 in orange. The figure was prepared using PyMOL (The PyMOL Molecular Graphics System v2.5.2 Enhanced for Mac OS X, Schrödinger, LLC).

**Figure 8 biology-12-00234-f008:**
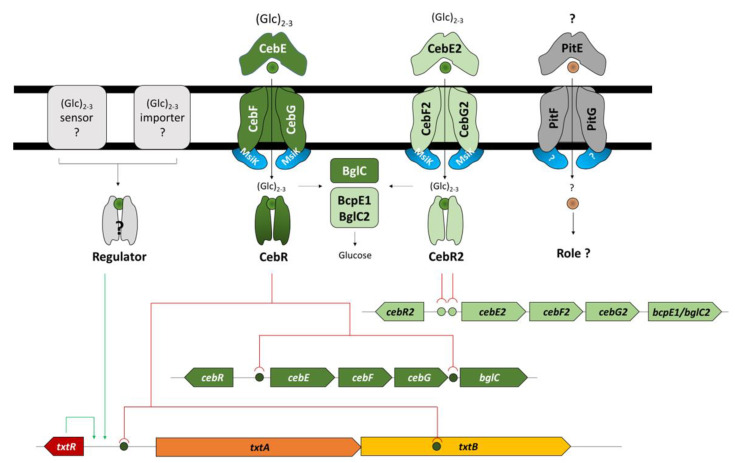
Model of cellobiose and cellotriose transport and thaxtomin production for the onset of pathogenicity in *S*. *scabiei* 87-22. Red lines, transcription repression; green lines, transcription activation.

## Data Availability

The data presented in this study are openly available. RNAseq data were publicly deposited, and our experimental and analytical pipelines are described in the GEO database repository (accession number: GSE181490). The CebE structure we used for modelization can be found at https://www.wwpdb.org/pdb?id=pdb_00008bfy accessed on 30 December 2022.

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
