# Peer review of "Role of Alternative Elicitor Transporters in the Onset of Plant Host Colonization by Streptomyces scabiei 87-22"

_biology, 2023, doi:10.3390/biology12020234_

Round 1

Reviewer 1 Report

The manuscript of Francis et al reports the existence and describes the role of a candidate protein complex CebEFG2 as an alternative route to CebEFG for the import of the virulence inducers cellobiose and cellotriose by S. scabies. The authors have previously identified the ABC transporter CebEFG-MsiK as the main importer of these sugars, which are the most efficient activators of S. scabies virulence in tested conditions. Cellobiose and cellotriose do indeed interact with TxtR and CebR, activating the expression of the thaxtomin phytotoxin biosynthetic genes. The authors also observed that the deletion of bglC in cebEFG didn’t abolish the survival on medium with cellobiose as the sole carbon source, indicating the existence of a compensating β-glucosidase feeding the glycolysis, named bglC2/bcpE1 in the cebEFG2 cluster. In this work, the authors test the hypothesis that this cebEFG2 cluster codes for an alternative cellobiose/cellotriose transport system. They begin by sharing the protein sequence homology-based research based on known cellobiose ABC-type transporter sequences of S. griseus and S. reticuli.  It led to the identification of CebEFG, CebEFG2 and PitEFG, a third pathogenicity island-situated transporter with no known role or substrate. CebEFG and CebEFG2 have a convincing sequence similarity and conserved organisation and CebEFG2 exists only in a subset of the S. scabiei harbouring CebEFG, making it a probable secondary transporter. Then, they rigorously prove that hypothesis by coupling expression level in the presence of cellobiose, obtaining deletion mutant and proceeding on growth assays on a minimal medium with cellobiose as sole carbon source, virulence assays on radish seedlings and thaxtomin HPLC quantification. I found it to be a well-written paper, scientifically sound. Thus, I recommend the acceptance of this manuscript provided that the comments and minor corrections listed below are addressed:

Line 105: “assays were prepared by growing the strains 104 as precultures in 5 ml TSB for 2 days at 28°C while shaking” Please indicate shaking speed in rpm.

Line 154: “The cluster with the highest homology to both CebEreti and CebEgris (scab57751-57721, Figure 1)” but in Figure 1 you indicate the positions scab57721-57761 for ceb1. Please choose the limits of the cluster and stick to them.

Lines 161-163: Are the members of these species not/less responding to cellobiose/cellotriose? Are they less virulent? What about S. scabies S58? and Streptomyces AMCC400023? Could you comment on that, please?

Lines 180-182: “Surprisingly, the 19 pathogenic strains that possess this putative second cello-oligosaccharide transporter all belong to the species S. scabiei” and no strains were discovered that had only ceb2. Could you comment on that, please?

Figure 2: Please indicate the bootstrap values.

More generally on the phylogenic analysis, it would be interesting to do a wider analysis of the conservation of cebEFG1 and cebEFG2 among the Streptomyces genus.

Figure 3: Induction of the levels of expression is stronger for the cellobiose than cellotriose but cellotriose is considered a more efficient inducer, could you comment on that point, please?

Line 257: “The double mutant for both cebE genes (ΔcebE1-2) was even more drastically impaired in growth on TDMc”. The growth delay looks slightly worse by the double mutant, maybe a better picture or a quantitative approach (e.g., counting of Colony Forming Unit) would help. If not, tone down “drastically”.

Figure 5: Using an 87-22 mutant non-producer of thaxtomins as negative control would ease the comparison between WT mutants and demonstrate the role of thaxtomins.

Lines 342-363: “Based on the crystallographic structures (…) are only partially conserved”. This part and Figure 7 don’t belong to the discussion part. Please, move it to the result section.

Line 596: Edit reference

Author Response

Line 105: “assays were prepared by growing the strains 104 as precultures in 5 ml TSB for 2 days at 28°C while shaking” Please indicate shaking speed in rpm.

Our response: The shaking speed has been specified.

Line 154: “The cluster with the highest homology to both CebEreti and CebEgris (scab57751-57721, Figure 1)” but in Figure 1 you indicate the positions scab57721-57761 for ceb1. Please choose the limits of the cluster and stick to them.

Our response: The position is now referred from scab57761 to 57721 in the text, as illustrated in Figure 1.

Lines 161-163: Are the members of these species not/less responding to cellobiose/cellotriose? Are they less virulent? What about S. scabies S58? and Streptomyces AMCC400023? Could you comment on that, please?

Our response: There is no or poor information on comparative virulence between the different pathogenic species. Moreover, we know that even strains that are phylogenetically clustered in the same group (like LBUM strains) display different virulence levels (while they have the same CebE background) (Hudec et al., 2021). It thus suggests that the virulence level is not just determined by the aa sequence of CebEFG or proteins involved in thaxtomin biosynthesis.

Lines 180-182: “Surprisingly, the 19 pathogenic strains that possess this putative second cello-oligosaccharide transporter all belong to the species S. scabiei” and no strains were discovered that had only ceb2. Could you comment on that, please?

Our response:  The ceb1 cluster is located in the core genome (middle of the linear chromosome) of S. scabiei. Hence, it is therefore plausible that its removal by chromosome deletion would severely affect the viability of S. scabiei strains while the loss of ceb2 that resides at the terminal part of the chromosome would have almost no consequences on viability.

Figure 2: Please indicate the bootstrap values.

Our response: Bootstrap values of the main nodes of the phylogenetic trees have now been included.

More generally on the phylogenic analysis, it would be interesting to do a wider analysis of the conservation of cebEFG1 and cebEFG2 among the Streptomyces genus.

Our response: Indeed, cebEFG2 is not only present in pathogenic streptomycetes. And it would be interesting also to include the other two cebEFG subgroups in the study, those orthologous to the cebEFG of S. griseus, and those orthologous to cebEFG of S. reticuli. Why do so many different CebEFG transport systems exist for the same function?  Is it only the affinity of the CebE protein for cellobiose and cello-oligosaccharides that differs between the clusters? Which clusters are found in highly cellulolytic species, species associated with the gut of termites, ...? There could be a logic for the CebEFG subgroups distribution. However, this goes beyond the scope of our manuscript.

Figure 3: Induction of the levels of expression is stronger for the cellobiose than cellotriose but cellotriose is considered a more efficient inducer, could you comment on that point, please?

Our response: As mentioned in Deflandre et al 2022, the metabolomic analysis revealed that cellotriose is the best elicitor for the production of specialized metabolites of S. scabiei including the thaxtomin phytotoxins. However, cellobiose is a better allosteric effector of CebR compared to cellotriose. It just means that, at the transcriptional level cellobiose is a better inducer of CebR-repressed genes, but in the end, the production levels of the final products are higher with cellotriose. This is probably because cellotriose is cleaved into cellobiose and glucose and therefore it activates expression at longer timepoints as it becomes cellobiose, the best ligand of CebR, once hydrolyzed by BglC. Cellobiose is instead hydrolyzed into two molecules of glucose and therefore does not possess a “second round” for preventing DNA-binding of CebR like cellotriose. Also, cellotriose is the best elicitor because it is the only molecule released from cellulose treated with thaxtomin. So, cellobiose is unlikely to be the natural elicitor.

Line 257: “The double mutant for both cebE genes (ΔcebE1-2) was even more drastically impaired in growth on TDMc”. The growth delay looks slightly worse by the double mutant, maybe a better picture or a quantitative approach (e.g., counting of Colony Forming Unit) would help. If not, tone down “drastically”.

Our response:  The word “drastically” has been removed.

Figure 5: Using an 87-22 mutant non-producer of thaxtomins as negative control would ease the comparison between WT mutants and demonstrate the role of thaxtomins.

Our response: It would indeed tell how big the role of thaxtomin is in the virulence response. We indeed demonstrated in Deflandre et al 2022 that not just thaxtomin production is induced by the elicitor(s) but also many other specialized metabolites that compose the virulome of S. scabiei 87-22. However, it is presumed that thaxtomin is one of the most important, if not the most important virulence factor produced by S. scabiei and our results on the thaxtomin levels extracted from the radish seedling assays reflect what is seen as the radish seedling phenotypes.

Lines 342-363: “Based on the crystallographic structures (…) are only partially conserved”. This part and Figure 7 don’t belong to the discussion part. Please, move it to the result section.

Our response: This point has been debated by the co-authors and we prefer to keep it in the discussion. Indeed, the structure modelization is used to discuss the possible differences in binding affinities between CebE1 and CebE2, but without biochemical data of CebE2 in complex with cellobiose and cellotriose, the text on the model structure of CebE2 consists of a series of hypotheses predicting the differences with CebE1. So, this is only speculation. We therefore prefer to keep it in the discussion.

Line 596: Edit reference

Our response: This issue has been fixed.

Reviewer 2 Report

Major comment:

S. scabies represents an original pathosystem responsible for a wide range of agronomic damages. The article submitted by Francis et al. deals with the identification and the characterisation of transporter systems in this bacterium that are paralogous to a previously described system involved in pathogenicity. Through a comprehensive set of experiments (in silico analyses, transcriptomics, plant tests...), the authors' results provide a better understanding of the signalling pathways of cello-oligosaccharide transport involved in the induction and regulation of the phytotoxin thaxtomin.

The paper is overall very well written, with original and robust data allowing a better understanding of the infectious mechanism of S. scabies. The authors' message is therefore of great interest to the scientific community and I am extremely supportive of the publication of this paper.

Questions/clarification points/corrections

Figure 1: As a non-specialist of S. scabies, I was surprised to see that the txt cluster was not part of the PAI (figure 1). If I’m not wrong, it is not mentioned in the text and I did confirm this information (Lerat et al. 2009). Maybe it would be possible to mention this fact somehow in the text?

L.163: I would change the formulation “instead contain the Ceb-reti” by something like “The CebE of pathogenic strains belonging to species XXX are phylogenetically related and grouped with the Ceb-reti”

L.178: I would avoid saying that ceb2 "has originated from one of the classical ceb clusters". While it is true that they are certainly homologous systems, it is difficult to infer a sense of evolution and say who is derived from whom.

Figure 2:  Bootstraps need to be added to support the different groups in the tree. It would for instance enable to really conclude whether all the CebE of the different species listed lines 160-161 are genuinely clustered with the CebE1 of the S scabies strains. I didn’t know the NGphylogeny plateform before, but bootstrap can be calculated in the advanced mode. The type of method used to build the tree could be also added (Distance/NJ, maximum likelihood …?). It would be nice also to include what the scale stands for.

In the figure legend, I didn’t really get what the authors mean by “only one representative of proteins with identical accession numbers” (L. 197).

L.223: LCF scab_77241+cellotriose=3.3 vs 3.19 in figure 3.

Figures 3/4 & L250-252: The authors stand that the Ceb2 cluster restore the growth of the cebEFG1 mutant. However, how can we explain that the cebEFG1 mutant is belated in its growth for 4 days if the cebEFG2 genes are already activated in transcriptomics from 1 H?

Figures 4/5. The authors deduce from their results that pitE and by extension the pitEFG cluster has no role in cello-oligosaccharide transport or plant colonisation. It is true that the pitE mutant does not show belated growth on plate or seem to change the pathogenicity of the strain, but in the ceb1-2 mutant, the pitE gene remains present and could be responsible for the belated growth of this mutant. Without having a ceb1/ceb2/pitE triple mutant, it seems to me difficult to exclude a role, indeed probably minor, of pitE in cello-oligosaccharide transport. Could not at least pitE be a potential candidate for the other transporter of cello-oligosaccharides (L395)?

Figure 6:  There are only standard deviations on the graphs. Could statistical tests be performed to assess differences between the different histograms? Only a mention in the legend that the observed differences are statically supported should be enough.

It is discussed L382-l386 that the nucleotide differences between cebR1 et cebR2 boxes could explain different transcriptional responses. But the presence of two cebR2 boxes (figure 1) could not be part of these transcriptional differences by dose effect (more binding of CebR)?

There are several double spaces in the text (L23, L221, L234, L407 etc…)

L197 : “CebE1 and and CebE2”remove one “and”.

Author Response

Figure 1: As a non-specialist of S. scabies, I was surprised to see that the txt cluster was not part of the PAI (figure 1). If I’m not wrong, it is not mentioned in the text and I did confirm this information (Lerat et al. 2009). Maybe it would be possible to mention this fact somehow in the text?

Our response: The genes pitEFG coding for the pathogenicity island-associated transporter are located in the CR (colonization region) of the split PAI of scabies, while the txt cluster is on the TR (toxicogenic region), more specifically TR1.

L.163: I would change the formulation “instead contain the Ceb-reti” by something like “The CebE of pathogenic strains belonging to species XXX are phylogenetically related and grouped with the Ceb-reti”

Our response: Modified as suggested.

L.178: I would avoid saying that ceb2 "has originated from one of the classical ceb clusters". While it is true that they are certainly homologous systems, it is difficult to infer a sense of evolution and say who is derived from whom.

Our response: The sentence was modified as follows “This high similarity between CebR (TGGGACGCGTCCCA) and CebR2 cis-acting elements further supports the hypothesis that these ceb clusters have originated from a common ancestor".

Figure 2:  Bootstraps need to be added to support the different groups in the tree. It would for instance enable to really conclude whether all the CebE of the different species listed lines 160-161 are genuinely clustered with the CebE1 of the S scabies strains. I didn’t know the NGphylogeny plateform before, but bootstrap can be calculated in the advanced mode. The type of method used to build the tree could be also added (Distance/NJ, maximum likelihood …?). It would be nice also to include what the scale stands for.

Our response: Bootstrap values of the main nodes of the phylogenetic trees have now been included.  All methods, parameters and software used were those set by default as described in (https://ngphylogeny.fr/workflows/advanced/fastme-oneclick?) with the exception of the option “Bootstrap branch supports” which was selected. The scale bar indicates the number of amino acid sequence substitutions per site.

In the figure legend, I didn’t really get what the authors mean by “only one representative of proteins with identical accession numbers” (L. 197).

Our response: A unique accession number is provided to proteins with identical amino acid sequences that originate from different organisms (See Table A2). Therefore, in order to make our phylogenetic trees readable, we decided to include only one representative protein for non-scabiei species when sometimes several strains of the same species have exactly the same sequence.  

L.223: LCF scab_77241+cellotriose=3.3 vs 3.19 in figure 3.

Our response: This typo has been corrected.

Figures 3/4 & L250-252: The authors stand that the Ceb2 cluster restore the growth of the cebEFG1 mutant. However, how can we explain that the cebEFG1 mutant is belated in its growth for 4 days if the cebEFG2 genes are already activated in transcriptomics from 1 H?

Our response: Even if the transcriptional response is immediate, we know nothing about the transport efficiency of the CebEFG2 transporter compared to CebEFG1. Maybe the KD of CebE2 for cellobiose is much higher (lower affinity) and therefore the transport efficiency would explain the growth delay. Also, it could simply be explained because in the WT you have 2 transporters for the same function (transport of cellobiose) while in the mutant only one is doing the job, and therefore less efficiently.

Figures 4/5. The authors deduce from their results that pitE and by extension the pitEFG cluster has no role in cello-oligosaccharide transport or plant colonisation. It is true that the pitE mutant does not show belated growth on plate or seem to change the pathogenicity of the strain, but in the ceb1-2 mutant, the pitE gene remains present and could be responsible for the belated growth of this mutant. Without having a ceb1/ceb2/pitE triple mutant, it seems to me difficult to exclude a role, indeed probably minor, of pitE in cello-oligosaccharide transport. Could not at least pitE be a potential candidate for the other transporter of cello-oligosaccharides (L395)?

Our response: It is correct what Reviewer 2 says: the pitE mutant does not show a growth delay as the cebEFG1 mutant does; this mutant also does not show a visual difference with the wild type when inoculated on radish seedlings; and in the cebE1-2 mutant this transporter should be unaltered in structure and function. It is indeed intriguing that the genes coding for this transporter are located on the PAI and hence are assumed to be important in virulence. Yet, under the conditions tested we did not observe a difference for the pitE mutant. This indeed does not exclude a role for this transporter in the S. scabiei pathology. However, a role in cellobiose/cellotriose transport seems unlikely based on the low sequence homology with identified CebEs and the absence of phenotypical difference with the wild type strain.

Figure 6:  There are only standard deviations on the graphs. Could statistical tests be performed to assess differences between the different histograms? Only a mention in the legend that the observed differences are statically supported should be enough.

Our response: A new figure is uploaded in the revised manuscript showing the statistical differences which indeed support the observations already made. This is also mentioned in the figure legend.

It is discussed L382-l386 that the nucleotide differences between cebR1 et cebR2 boxes could explain different transcriptional responses. But the presence of two cebR2 boxes (figure 1) could not be part of these transcriptional differences by dose effect (more binding of CebR)?

Our response: Yes, it is possible that doubling the boxes is part of the molecular mechanism in place for the transcriptional control. But probably what is also important, and not known, is the concentration level at which cellobiose and cellotriose can act as allosteric effectors of CebR. This is more important than doubling the cis-element. For instance, many cellulase encoding genes have multiple boxes but show the best transcriptional response to either cellobiose or cellotriose. The position of the CebR-binding sites according to the –10 and –35 boxes is also of major importance.

There are several double spaces in the text (L23, L221, L234, L407 etc…)

Our response:  The double spaces have been tracked and corrected.

L197 : “CebE1 and and CebE2”remove one “and”.

Our response: This has been corrected.

Reviewer 3 Report

The authors investigated potential role of Streptomyces scabiei's cello-oligosaccharide transporters in activation of elicitors affecting bacterial virulence. The manuscript is informative and well-written. The methodologies used have yielded satisfactory results to draw a conclusion and set foundation for the next phases of research. The paper could be considered for publication after addressing the minor comments below.

1) Page (P) 1, Line (L) 11-13, rephrase the sentence to clarify the meaning.

2) L16, spell out ABC transporter as the first use.

3) L25, add cebEFG1 after transporter to make sure what the main means.

4) P2, L35-37, this part needs to be rephrased.

5) L42, L73, since the genus is the same, only use them as 'S' in the beginning after the first use. Just use S. reticuli.

6) P3, L91, spell out PAI a first use.

7) P6, L200, although, there are some methodologies of this section presented in the caption of figure 3, make sure to add such information in the materials and methods as well.

8) P7, L229-231, this section belongs to the discussion part.

9) P11, L344, remove the website; it's been mentioned once in the materials and methods section.

10) L346-364, this paragraph belongs to the results section; revise the figure 7 in the text as well accordingly.

Author Response

1) Page (P) 1, Line (L) 11-13, rephrase the sentence to clarify the meaning.

Our response: To better clarify its meaning, the sentence has been changed to: “Our work investigated if and how this pathogen uses multiple sugar transport systems to sense the presence of a living plant host through perception and import of cello-oligosaccharide, the elicitors that trigger the onset of the pathogenic lifestyle of S. scabiei.”

2) L16, spell out ABC transporter as the first use.

Our response: Modified as requested.

3) L25, add cebEFG1 after transporter to make sure what the main means.

Our response: This has been added.

4) P2, L35-37, this part needs to be rephrased.

Our response: It is rather difficult to rephrase this sentence as the Reviewer does not point out what the actual problem with the sentence is. Hence, we decided to leave the sentence as the other two Reviewers did not seem to have a problem with it. If a better clarification can be given on what the Reviewer would like to see as change or point out what the problem is with this sentence, we would gladly abide by the Reviewer’s request.

5) L42, L73, since the genus is the same, only use them as 'S' in the beginning after the first use. Just use S. reticuli.

Our response: Modified as requested.

6) P3, L91, spell out PAI a first use.

Our response: Modified as requested.

7) P6, L200, although, there are some methodologies of this section presented in the caption of figure 3, make sure to add such information in the materials and methods as well.

Our response: All necessary information has now been included in the materials and methods section.

8) P7, L229-231, this section belongs to the discussion part.

Our response: We discussed this with the co-authors, and we feel that at this point in the manuscript we have to mention that our data align with previous results to show that we are aware of the previous work. We also briefly added this information in the discussion section.

9) P11, L344, remove the website; it's been mentioned once in the materials and methods section.

Our response: The website link has been removed.

10) L346-364, this paragraph belongs to the results section; revise the figure 7 in the text as well accordingly.

Our response: This has been debated between co-authors and we prefer to keep it in the discussion. Indeed, the structure modelization is used to discuss the possible differences in binding affinities between CebE1 and CebE2, but without biochemical data of CebE2 in complex with cellobiose and cellotriose, the text on the model structure of CebE2 consists of a series of hypotheses predicting the differences with CebE1. So, this is only speculation. We therefore prefer to keep it in the discussion.